# Hydrotalcite–Niclosamide Nanohybrid as Oral Formulation towards SARS-CoV-2 Viral Infections

**DOI:** 10.3390/ph14050486

**Published:** 2021-05-19

**Authors:** Goeun Choi, Huiyan Piao, N. Sanoj Rejinold, Seungjin Yu, Ki-yeok Kim, Geun-woo Jin, Jin-Ho Choy

**Affiliations:** 1Intelligent Nanohybrid Materials Laboratory (INML), Institute of Tissue Regeneration Engineering (ITREN), Dankook University, Cheonan 31116, Korea; goeun.choi@dankook.ac.kr (G.C.); 12192032@dankook.ac.kr (H.P.); sanojrejinold@dankook.ac.kr (N.S.R.); 32182808@dankook.ac.kr (S.Y.); 2College of Science and Technology, Dankook University, Cheonan 31116, Korea; 3Department of Nanobiomedical Science and BK21 PLUS NBM Global Research Center for Regenerative Medicine, Dankook University, Cheonan 31116, Korea; 4Department of Chemistry, College of Science and Technology, Dankook University, Cheonan 31116, Korea; 5R&D Center, CnPharm Co., Ltd., Seoul 03759, Korea; kyeoksky@cnpharm.co.kr; 6Department of Pre-medical Course, College of Medicine, Dankook University, Cheonan 31116, Korea; 7Tokyo Tech World Research Hub Initiative (WRHI), Institute of Innovative Research, Tokyo Institute of Technology, Yokohama 226-8503, Japan

**Keywords:** niclosamide, hydrotalcite, nanohybrid, oral formulation, COVID-19, pharmacokinetics

## Abstract

COVID-19 has been affecting millions of individuals worldwide and, thus far, there is no accurate therapeutic strategy. This critical situation necessitates novel formulations for already existing, FDA approved, but poorly absorbable drug candidates, such as niclosamide (NIC), which is of great relevance. In this context, we have rationally designed NIC-loaded hydrotalcite composite nanohybrids, which were further coated with Tween 60 or hydroxypropyl methyl cellulose (HPMC), and characterized them in vitro. The optimized nanohybrids showed particle sizes <300 nm and were orally administrated to rats to determine whether they could retain an optimum plasma therapeutic concentration of NIC that would be effective for treating COVID-19. The pharmacokinetic (PK) results clearly indicated that hydrotalcite-based NIC formulations could be highly potential options for treating the ongoing pandemic and we are on our way to understanding the in vivo anti-viral efficacy sooner. It is worth mentioning that hydrotalcite–NIC nanohybrids maintained a therapeutic NIC level, even above the required IC_50_ value, after just a single administration in 8–12 h. In conclusion, we were very successfully able to develop a NIC oral formulation by immobilizing with hydrotalcite nanoparticles, which were further coated with Tween 60 or HPMC, in order to enhance their emulsification in the gastrointestinal tract.

## 1. Introduction

COVID-19 has been affecting millions of individuals globally, leaving lives miserable, especially in under-developed countries and even in poor areas of cities in advanced countries [1]. COVID-19 is typically affected by SARS-CoV-2 virus, and is highly transmissible compared to previous variants, such as MERS and SARS [2,3,4], which previously affected East Asian countries in the early 2000s. [5] Though the number of cases are increasing all around the world, and vaccinations have started globally, actual therapy for COVID-19 remains onerous to many scientists. On the other hand, the present pandemic situation is severe, which requires further development of urgent medications as soon as possible.

There have been various drug candidates that have showed anti-viral potency against SARS-CoV-2, among which niclosamide (NIC) is of great importance. Interestingly, Wu et al. [6] and Gassen et al. [7] reported that NIC can inhibit the viral replication of SARS-CoV and MERS-CoV, respectively. In fact, a very recent study revealed that NIC had a strong anti-viral activity against SARS-CoV-2 with a reported IC_50_ and IC_100_ of 0.28 µM (~100 ng/mL) and 1 µM (~300 ng/mL) [8]. These in vitro experimental results highlighted how efficacious NIC would have been if one could have successfully engineered it in a suitable drug carrier to improve its well-known poor bioavailability.

Although NIC showed significantly higher anti-viral activity in vitro, in vivo applicability is highly challenging, mainly because of its poor aqueous solubility, making it inadmissible for actual clinical applications. For example, clinical trial NCT02532114 was terminated due to the low oral bioavailability of NIC. According to the study, the C_max_ value was lower than 200 ng/mL, even after oral administration in patients three times/day (500 mg) [9].

It is worth mentioning that nanotechnology has advanced dramatically in terms of nanomedicine research areas, and, particularly, in drug delivery with the aid of functionalized organic, inorganic, and hybridized particles [10,11,12,13,14,15,16,17,18,19,20,21]. Such nanocarriers are expected to enhance overall solubility, thereby improving the clinical outcomes of NIC. Polymer-based formulations, such as hydroxy-propyl-β-cyclodextrin [22] and chitosan [23], have also been previously explored for improving the therapeutic benefits of NIC. However, the pharmaco-kinetic (PK) parameters showed that there were no significant improvements with these formulations. Therefore, inorganic-based nano-structures could be an alternative to enhance the overall action of the poorly soluble NIC drug. Among the inorganic nanomaterials, clay-based nanoparticles (NPs), in particular layered double hydroxide (LDH) [24] and montmorillonite (MMT) [25,26,27,28,29,30,31,32,33,34], have been widely explored for nanomedical applications owing to their excellent biocompatibility, biodegradability, and stimuli-responsive characteristics [35,36].

We have also extensively studied LDH as a smart drug-delivery carrier for the enhancement of drug stability, bioavailability, and solubility of poorly soluble drugs, such as artesunic acid, ursodeoxycholic acid, donepenzil, indole-3-acetic acid, etc. [35,37,38,39]. LDHs belong to a class of anionic clays. The general formula of LDH is [M(II)_1-x_ M(III)_x_ (OH)_2_]^x+^[A^m-^]_x/m_·nH_2_O, where M(II) is a divalent cation, M(III) is an isomorphously substituted trivalent one, and A^m-^ is an interlayer anion. In particular, LDH with Mg^2+^ and Al^3+^ in its octahedral site and CO_3_^2−^ in the interlayer is known as hydrotalcite (HT; Mg_6_A_l2_(OH)_16_CO_3_·4H_2_O). In addition, by heating beyond ~450 °C, HTs can be converted into an X-ray amorphous metal oxide with Lewis basic and acidic properties and non-swelling behavior, which make them valuable for a wide range of applications [40,41,42,43].

The important reason for selecting HT as a nanocarrier was due to its well-known biocompatibility, according to our previous studies on developing drug delivery systems [44,45,46]. In addition, its muco-adhesion property is expected to be beneficial as well.

In general, anti-viral agents can work through extra-cellular and intra-cellular pathways; however, in both cases, their mechanisms are different. NIC has three major mechanisms, (i) inhibiting viral interaction with the hACE2 (human angiotensin converting enzyme-2) receptor; (ii) inhibiting viral replication; and (iii) blocking its autophagy, as shown in Scheme 1.

In the case of SARS-CoV-2, with a size in the range of 50–150 nm, it can easily enter into human cells, where it rapidly replicates and matures, potentially leading to fatal consequences. In this context, it is necessary to engineer the drug delivery carrier at a nano scale in order to effectively tackle the virions. Additionally, oral administration of such well-engineered nanoparticles (NPs) would be therapeutically efficient, as the viral loads are known to be higher in the gastrointestinal tract (GIT) [47]. However, NIC has been reported to undergo very fast elimination via the hepatic metabolism as NIC-glucuronic acid [48], and this will reduce a large amount of orally administered drugs. If one could strategically immobilize NIC in a suitable drug delivery carrier, this fast metabolism of NIC could be altered, resulting in an improvement in NIC solubility, and eventually in in vivo efficacy.

In the present study, an attempt was made to increase the specific surface area of HTs in order to improve the loading capability of NIC molecules by calcining around 300 °C. In this way, the interlayer and physisorbed H_2_O in the two-dimensional HT lattice was removed and the formation of dehydrated HT (DHT), having a higher surface area compared to intact HT, was achieved. Thereafter, the NIC molecules were loaded as prepared DHT to finally generate NIC–DHT nanohybrids with drug content of ~43.3 % (Scheme 2a), which was further coated with a muco-adhesive non-ionic polymer, either Tween 60 or hydroxypropyl methyl cellulose (HPMC), as shown in Scheme 2b, to improve the overall solubility in vivo, thereby resulting in a high bioavailability. Such rationally designed NIC–DHT/Tween 60 and NIC–DHT/HPMC nanohybrids would be ideal for imparting high in vitro and in vivo anti-viral effects, and are under investigation with the aim of providing a repurposed NIC formulation for real life applications against COVID-19 in the very near future.

## 2. Results and Discussion

### 2.1. Powder X-ray Diffraction (PXRD) Analysis

In order to evaluate the crystallinity of NIC, HT, DHT, and NIC–DHT nanohybrids, we performed powder X-ray diffraction (PXRD) analyses (Figure 1). The 2D characteristic peaks of (*00l*) for HT were seen at 11.6° for (*003*), and 23.3° for (*006*), respectively (Figure 1b), which were transformed into pseudo-3D broad peaks after calcining at around 300 °C (Figure 1c) due to the formation of nonstoichiometric periclase (MgO) containing Al^3+^ ions [49]. After hybridizing DHT with NIC, the characteristic XRD peaks for DHT and NIC remained unchanged (Figure 1d).

The obvious changes, such as being fairly amorphous and having a pseudo-3D structure, were well confirmed by PXRD analyses, as mentioned above, indicating that NIC could be efficiently loaded and protected to impart better water solubility. This might be useful in enhancing in vitro/in vivo anti-viral efficacy in our future studies. There have been several reports on clay-based nanomaterials showing improved water solubility for poorly soluble drugs, such as curcumin and NIC. For example, a montmorillonite (MMT)-based drug delivery system showed sustained curcumin [50] and NIC [51] release when encapsulated in MMT. Similarly, our well-engineered NIC–DHT nanohybrids might substantially improve NIC therapeutic outcomes. Furthermore, the main reasons for coating non-ionic polymers over NIC–DHT were, not only to stabilize the NIC–DHT nanohybrid in the GIT, but also to enhance eventual NIC release. Formulation with non-ionic polymers in this study was not subjected to chemical modification with NIC–DHT, only a simple physical mixing was performed in order to uniformly coat the non-ionic polymers over NIC–DHT (Appendix A). UV analyses clearly showed that there was no peak shift, even after Tween 60 or HPMC coating, indicating that the non-ionic polymer was simply coated on the NIC–DHT surface by physical adsorption (Appendix A).

### 2.2. Fourier Transform Infrared (FT-IR) Analysis

Figure 2 shows the Fourier transform infrared (FT-IR) spectra of intact NIC, HT, DHT and NIC–DHT nanohybrids. The intact NIC showed major characteristic peaks at 3578 cm^−1^, 3490 cm^−1^, 1680 cm^−1^, 1510 cm^−1^, and 540 cm^−1^, and could be assigned as -OH, -NH, -C = O, -NO_2_, and C–Cl groups, respectively (Figure 2a) [51]. For pristine HT, broad bands around at 3400 cm^−1^ and at 1630 and 1545 cm^−1^ were observed, which can be ascribed to (O–H) group vibrations owing to both the hydroxide layer and the interlayered water (Figure 2b). On the other hand, the characteristic –OH and the –CO_3_^2−^ bands were fairly reduced after the calcination process, which was evidenced by TGA (thermogravimetric analysis) and DTA (differential thermal analysis), as shown in Appendix A. Interestingly, the carbonate peak at ~1360 cm^-1^ corresponding to HT was split in two (Figure 2c) indicating their structural transformation to DHT, as previously reported by our group [52]. One thing to note here was that the specific bands for –OH and –NH were reduced in the NIC–DHT sample after hybridization, indicating the successful loading of NIC on DHT.

### 2.3. Surface Areas and Porosity Analysis

The nitrogen adsorption–desorption isotherms of HT, DHT and NIC–DHT nanohybrids are shown in Figure 3. The values for the respective surface areas (S_BET_) and total pore volumes (V_p_) were calculated from the adsorption isotherms using the BET method (Table 1). The calcined DHT samples exhibited a higher surface area (62.18 m^2^/g) than that of uncalcined ones (9.05 m^2^/g), which was attributed to its concave–convex type of surface. Accordingly, the corresponding total pore volume was increased from 0.061 to 0.099 cm^3^/g. In contrast, the surface area (6.85 m^2^/g) and pore volume (0.035 cm^3^/g) of the NIC–DHT nanohybrid was lower than that of the DHT sample, indicating that the NIC molecules must have immobilized on DHT which has an irregular surface pattern.

As we can see clearly from the N_2_ adsorption–desorption results (Figure 3), the specific surface areas (S_BET_) for both samples, HT and NIC–DHT, were very low and almost negligible compared to that for porous DHT. This is evidence that the pores in the DHT were filled with NIC drug molecules to form NIC–DHT, resulting in the same adsorption-desorption curves as HT.

### 2.4. FE-SEM Analysis

The electron microscope characterizations of HT, DHT, and NIC–DHT nanohybrids are shown in Figure 4. FE-SEM images show the step-by-step morphologies from HT to NIC–DHT nanohybrids (Figure 4). The pristine HT exhibited a plate-like morphology with a smooth particle surface and a diameter of ∼300 nm, which was typical for a layered material. After calcination of HT, the average particle size and morphology of DHT were mostly maintained, but the surface became slightly rough compared to that of HT, which might be due to the dehydration and partial decarbonation of HT during calcination [53]. On the other hand, the morphology of NIC-DTH was retained, while the surface became irregular, compared to both HT and DHT plates, due to the adsorption of NIC molecules on the surface.

### 2.5. Particle Size Analysis

According to dynamic light scattering (DLS) analysis, the average particle size of pristine HT, DHT, and NIC–DHT nanohybrids were comparable to each other, with average values of 279.9 ± 35.6, 268.5 ± 23.8, and 292.7 ± 28.3 nm (Figure 5), respectively, which were in good agreement with the FE-SEM images. The polydispersity index (PDI) values of pristine HT, DHT, and NIC–DHT nanohybrids were 0.286, 0.297, and 0.139, respectively, indicating that they were all well dispersible since all the PDI values fell under the range of 0.3.

Since particles were in the optimum range of <300 nm (as per DLS and FE-SEM), they could be well used as an ideal anti-viral therapeutic agent. This is because the SARS-CoV-2 virus has smaller particle size (50–150 nm) and, therefore, our engineered NIC–DHT nanohybrids should be able to enter virus-infected cells and induce anti-viral effects. Our research group previously established the potential endocytosis mechanism involved in LDHs [54]. Therefore, it is important to protect readily clearable drug candidates, such as NIC, well if they are administered orally and parenterally. In most reported cases, NIC has shown significantly lower plasma concentrations after oral administration [55]. Therefore, protecting them in an ideal nanocarrier, such as DHT, would be beneficial to achieve better therapeutic NIC levels on oral or parenteral administration.

### 2.6. NIC Contents in the Nanohybrids

A drug content analysis is given in Section 4.5 and the Supporting Information. NIC–DHT hybrid was dissolved in an HPLC mobile phase (acetonitrile:10 mM ammonium acetate (0.1% formic acid) = 80:20, v/v) and filtered with a nylon syringe filter (0.22 um). Subsequently, the amount of NIC loaded in the hybrid was determined from a calibration curve obtained from a series of NIC solutions in different concentrations. The determined NIC contents in NIC–DHT, NIC–DHT/Tween 60, and NIC–DHT/HPMC nanohybrid formulations were 43.3 ± 1.5 %, 33.5 ± 0.3 %, and 29.3 ± 1.8 %, respectively. A reduced drug content was obtained for NIC–DHT/Tween 60 and NIC–DHT/HPMC nanohybrid formulations compared to the NIC–DHT nanohybrid which could be due to the surface coating of non-ionic polymers (Tween 60 and HPMC).

### 2.7. In Vivo Pharmacokinetics of NIC

An in vivo PK study was conducted for NIC nanohybrids after a single oral administration using rats, to obtain information on plasma drug concentrations. The in vivo PK study was done using NIC oral formulations, such as NIC–DHT/Tween 60 and NIC–DHT/HPMC nanohybrids. A higher oral dosage of 200 mg/kg confirmed the feasibility of the formulation to be translated in vivo.

We compared NIC concentration profiles in plasma after oral administration of NIC–DHT/Tween 60 with that of NIC–DHT/HPMC for up to 24 h. The curves of the NIC plasma concentration versus time in rats are shown in Figure 6. It should be noted that NIC plasma concentration at 0.25 h of NIC–DHT/Tween 60 (50 mg/kg) nanohybrids was around 1350.37 ± 613.98 ng/mL, which is higher than the highest NIC plasma concentration after oral administration of Yomesan^®^ (4 h) (Appendix A) [51]. This clearly indicates improved PK effects of NIC once immobilized on DHT nanohybrids. In addition, the fast systemic circulation of NIC might be beneficial for an effective therapeutic strategy against SARS-CoV-2 virus, especially in the initial symptomatic and asymptomatic phases.

PK profiles were subsequently optimized by changing the dosage and the type of coated polymer. Upon changing the dosage from 50 to 200 mg/kg for NIC–DHT/Tween 60, the plasma concentration of NIC above an IC_50_ value was maintained up to 8 h, but ~4.7-fold increase of NIC plasma concentration was observed at 0.25 h. To further optimize the PK profile, the NIC–DHT surface was coated with HPMC, which has a better gastrointestinal retentive property than Tween 60. The PK profile acquired after a single oral administration of NIC–DHT/HPMC showed that the NIC plasma concentration above IC_50_ was sustained beyond 24 h. As far as the IC_100_ value was concerned, NIC plasma concentration was also significantly prolonged for 12 h.

We hypothesize that the muco-adhesive properties of HT and the non-ionic polymers (Tween 60 and HPMC) led to the improved PK profiles. Another hypothesis is that the enhancement of bioavailability could have been caused by evading or altering the fast intestinal or hepatic metabolism via cytochrome-P450 enzymes. It has already been reported that NIC could undergo fast metabolism in liver and this would eliminate most of the orally administered drug into NIC-glucuronic acid [48]. It should be highlighted that our rational molecular engineering strategy using DHT and non-ionic polymers with NIC became successful by enhancing the muco-adhesive property, and thereby, NIC could be sustained in the intestinal track and metabolized through hepatic pathway. Based on the present strategy we were able to achieve high plasma concentration of NIC after a single oral administration.

To the best of our knowledge, the present work is the first describing the pharmacokinetics of oral NIC formulations sustaining plasma concentration above IC_50_ for 24 h. Further, when considering the medical applications of this formulation, we were able to achieve a maximum therapeutic NIC concentration, even above the IC_100_, for 12 h, assuming that most of the orally administered drug has entered into systemic circulation. The as-made NIC–DHT/HPMC nanohybrid was found to sustain this therapeutic concentration up to 24 h in plasma (Figure 6 and Table 2), which would be highly useful for combating the initial symptomatic and asymptomatic phases, very commonly found conditions in COVID-19 patients.

## 3. Clinical Perspectives

Since COVD-19 is a rapidly evolving emergency situation, there has been a tremendous amount of attention given to formulating different medical strategies in terms of early diagnosis, detection, prevention, and therapy. It is worth mentioning that nanotechnology plays a vital role in all the dramatic and unbelievable progress in science [56,57,58,59,60,61,62,63]. Using wide variety of functional nanomaterials, it is possible to specifically target pathogens, thereby improving the overall efficacy of therapeutic outcomes.

In our study, the major challenge was to improve the in vivo PK parameters by safely loading NIC onto DHT nanohybrids. Why this is so important for COVID-19 therapy is the key question we focused on throughout our experimental studies. In fact, NIC has been well known for its anti-viral efficacy on a wide variety of viral infections, as was reviewed by Xu et al. (2020) [64]. Nevertheless, the major limitation associated with its poor water solubility has been an ongoing challenging issue for pharmacologists.

Recent studies showed that clay-based nanomaterials could selectively target SARS-CoV-2 virus, though the mechanisms were unclear [65]. In this context, we were successful in loading NIC onto DHT nanohybrids, which are also clay-based NPs. Since our NIC–DHT nanohybrid had an optimum particle size <300 nm, we believed that it penetrate into infected cells, thereby inhibiting RNA replication and stopping further maturation in a controlled way.

The highlight of our in vivo PK study was an improvement of the AUC and T_1/2_ for NIC from the present nanohybrid drug compared to that of the commercially available Yomesan^®^ [51], with a single oral dosage. This kind of rational design strategy is important as the proof of concept in our present study, to have a sustained drug release in the GIT in a way that achieves an enhanced bioavailability for poorly soluble drugs, such as NIC. Our future studies will focus on understanding how orally administered NIC–DHT/Tween 60 and NIC–DHT/HPMC are able to significantly enhance NIC plasma concentrations; how DHT plays a key role in improving such oral bioavailability for NIC; and if these NIC nanohybrids are able to retain very high therapeutic efficacy in in vitro and in vivo models. The answers to these questions will definitely be able to translate our present bench work into clinical trials, hoping that the present molecularly engineered NIC nanohybrids could potentially show anti-COVID-19 effects.

## 4. Materials and Methods

### 4.1. Materials

Niclosamide was purchased from DERIVADOS QUIMICOS. Trifluoracetic acid and ethanol were acquired from Daejung Chemicals. Hydrotalcite (HT; Mg_6_Al_2_(CO_3_)(OH)_16_ 4H_2_O) and Tween 60 were purchased from Sigma-Aldrich (Korea) and TCI (Japan), respectively. HPMC (6 mPa∙s) was kindly donated by Wonpoong Pharm Co., Ltd.

### 4.2. Preparation of DHT, NIC–DHT and Tween 60 (or HPMC)-Coated NIC–DHT

The HT was completely dried in furnace (~300 °C) for 8 h to produce DHT. DHT (3.0 g) was mixed with NIC (1.0 g) and dissolved in ethanol. The mixture was stirred for 6 h, washed with ethanol and then filtered to prepare the final NIC–DHT nanohybrids. For the preparation of Tween 60-coated NIC–DHT, the NIC–DHT powder (2025 mg) was mixed with Tween 60 (795 mg), and the samples were dispersed in 10 mL ethanol and dried using a rotary evaporator. HPMC-coated NIC–DHT was prepared by mixing NIC–DHT powder (2025 mg) with HPMC (395 mg) and subsequently dispersing it in 10 mL ethanol. The mixture was dried using a rotary evaporator.

### 4.3. TGA

The elimination of H_2_O and CO_3_^2−^ during the calcination process of HT was confirmed by TGA and DTA analyses (Appendix A). The experiment was conducted using a TA instrument (SDT Q600, Universal Analysis2000 (New Castle, DE, USA)) with a sample size of 15 mg (scraped from the coated specimen) at a heating rate of 20 °C/min to 800 °C under flowing air (10 mL/min)

### 4.4. Characterization of NIC–DHT Hybrid

The PXRD patterns for the NIC, HT, DHT, and NIC–DHT hybrids were obtained with a Bruker D2 Phase diffractometer (Bruker, Karlsruhe, Germany) equipped with Cu K_α_ radiation (λ = 1.5418 Å). All the data were recorded at the tube voltage and a current of 30 kV and 10 mA, respectively. The UV–Visible absorption for the NIC, HT, DHT, NIC–DHT, NIC–DHT/Tween 60, and NIC–DHT/HPMC were measured in EtOH (99.9%) at 333 nm by a Jasco UV/Vis spectrometer (V-630, Easton, MA, USA). The FT-IR spectra were recorded with a Jasco FT/IR-6100 spectrometer (JASCO, Tokyo, Japan) using the standard KBr disk method in transmission mode (spectral range 4000–400 cm^−1^, resolution 1 cm^−1^, 40 scans per spectrum). The morphology analysis of different samples, such as NIC, HT, DHT, and NIC–DHT hybrids, were studied by using a Sigma 300 (Carl Zeiss, Oberkochen, Germany) field-emission scanning electron microscope. The particle sizes of the HT, DHT, and NIC–DHT hybrids were measured by using a particle size analyzer (ELSZ-2000ZS; Otsuka, Tokyo, Japan) in EtOH (99.9%) as the solvent. In addition, the zeta potential evaluation of DHT, NIC–DHT, NIC–DHT/Tween 60, and NIC–DHT/HPMC were measured by using zeta potentiometer (ELSZ-2000ZS; Otsuka, Tokyo, Japan) in distilled water containing 0.1% Tween 60 (Appendix A). All measurements were done in triplicates (*n* = 3). Nitrogen physisorption measurements were performed at 77 K on a BELSORP II mini instrument (BEL Japan, Inc., Osaka, Japan). The HT and DHT were degassed for 6 h at 100 °C before measurements, whereas the NIC–DHT hybrid was degassed for 12 h at room temperature.

### 4.5. High Performance Liquid Chromatography

The samples were measured at 330 nm using Agilent HPLC system with a Poroshell 120 column (21 × 100 mm, 2.7 µm) (Agilent, Santa Clara, California, USA) at a 0.4 mL/min flow rate. An isocratic method was used with a mobile phase consisting of acetonitrile and 10 mM ammonium acetate (0.1% Formic acid) in an 80:20 ratio.

### 4.6. Pharmacokinetic Study in Rats

The PK study was performed in male Sprague-Dawley rats (7-week-old, 232–255 g). The SD rats were housed in a light controlled room, where the temperature and relative humidity were maintained at 23 ± 3 °C and 50 ± 5%, respectively. The animal study was approved by the KNOTUS Institutional Animal Care and Use Committee (KNOTUS IACUC; Protocol Number: 21-KE-110). Prior to the animal experiments, NIC–DHT/Tween 60 (50 mg/kg) was dispersed in water and NIC–DHT/Tween 60 (or HPMC) (200 mg/kg) was dispersed in 5% Tween 60/water solution. For comparison of niclosamide concentration profiles in plasma after oral administration of formulations, we observed the profiles up to 24 h. In this study, it was not possible to collect blood more than 10 times from rats in 24 h since excessive blood collection resulted in the death of the animal model. For blood sampling, approximately 0.25 mL of blood was collected from each of the rats at 0, 0.25, 0.5, 1, 2, 4, 6, 8, 12, and 24 h after oral administration of NIC–DHT/Tween 60 (Group 1: 50 mg/kg, Group 2: 200 mg/kg, *n* = 5 for each group) or NIC–DHT/HPMC nanohybrid (Group 3: 200 mg/kg, *n* = 5). The blood samples were centrifuged for 2 min at 13,000 rpm in order to separate the plasma which was immediately frozen until further analysis by HPLC-mass spectrometry (HPLC-MS; Waters, Milford, MA, USA).

### 4.7. Quantification of Niclosamide in Rat’s Plasma

A plasma calibration curve was prepared by using NIC standards at concentrations of 0, 5, 10, 50, 100, 500, 1000, and 2000 ng/mL. Samples were prepared by adding 100 µL of internal standard, topimerate, to 20 μL aliquots of rat plasma sample. Then, they were vortexed for 5 seconds, followed by centrifugation at 13,000 g for 5 min at 4 °C. The 20 µL of supernatant was mixed with 180 µL of 50% methanol. Then, 150 µL of the aliquots of the samples were transferred to sample vials for the analyses. Samples were analyzed using an Acquity I-Class UPLC system (Waters, Milford, MA, USA) with a mass spectrometer (Waters Xevo TQ-S, Milford, MA, USA). The LC analytical column was a Thermo Hypersil Gold (2.1 x 50 mm, 1.9 μm) and was maintained at 50 °C during the analyses. The mobile phase was prepared with 5 mM ammonium acetate water solution and methanol with a flow rate of 0.4 mL/min.

## 5. Conclusions

In conclusion, we successfully developed NIC–DHT/Tween 60 and NIC–DHT/HPMC nanohybrid oral formulations with enhanced oral bioavailability of poorly water soluble NIC drug. This is the first attempt in utilizing HT as potential oral drug delivery agent for NIC drug delivery. The as-made NIC–DHT nanohybrids showed optimum particle sizes <300 nm with excellent PK profiles after coating with Tween 60 or HPMC. The oral formulation of NIC–DHT/Tween 60 and NIC–DHT/HPMC was realized on the basis of the hypothesis that HT and non-ionic coating polymers could improve PK profiles of NIC due to their muco-adhesive property. As expected, the NIC plasma levels were maintained above IC_50_, required for maintaining drug efficacy, for 24 h, with a single administration. As a proof of the concept, our well-designed oral formulation of NIC–DHT coated with Tween 60 or HPMC could be a potential anti-viral agent to combat the ongoing pandemic. Further studies are on the way to understand the in vivo anti-viral efficacy of the present formulation. We hope and believe that DHT-based nanomaterials, encapsulating high potential FDA-approved anti-viral drugs, could selectively target many pathogens. Therefore, we are looking forward to understanding their specific applications towards a wide variety of diseases that have been extremely challenging to tackle.

## Data Availability

The data presented in this research study are available in this article.

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
