# Peer review of "Hydrotalcite–Niclosamide Nanohybrid as Oral Formulation towards SARS-CoV-2 Viral Infections"

_pharmaceuticals, 2021, doi:10.3390/ph14050486_

Round 1
Reviewer 1 Report
The manuscript deals with a pharmaceutical formulation with perspective in therapy. Still, the use of Niclosamide on Sars-Cov-2 infection is not yet approved. Thus, I suggest the modification of the title of the paper.
The author formulated a new pharmaceutical form based on Niclosamide (NIC), an antihelminthic drug with potential antiviral effect according to recent literature. The authors sustain this research as a future therapy for Sars-Cov-2 infection. The pharmaceutical form is a drug deliver system for an active substance which assure an increased absorption of the drug and it has perspective for increasing the bioavailability not only for NIC but also could be a solution for other molecules.
The research is interesting from the point of view of pharmaceutical technology. Still, niclosamide is not yet approved for the therapy of COVID-19 and this is why I suggested the modification of the title.
The paper is original because the authors propose a new pharmaceutical form, they characterize it and sustain an improved absorption of the active molecule.
The paper is well explained and easy to understand.
The conclusions are according to the data presented and also the authors mention the future studies which should be performed to support the use in COVID-19 therapy.
I recommend the publication of the paper because there is an improved solution for drug-delivery through the intestinal membrane and it could be of future interest. The paper presents sufficient data for publication.
Still, there is a suspicion on the real effect on Sars-Cov-2 infection and the systemic toxicity of Niclosamide which could diminish the therapeutic use. The authors consider it for future studies. So, the manuscript could be also considered for rejection because of this issue.
Reviewer 2 Report
It is important that scientists from all over the world are working together in the fight against COVID-19 infection. We are therefore delighted to be able to review the work you have done.
Now, I have a few questions, and I would like to ask you to add your comments in a polite and refutable way.Basically, I admire your work.
(My English grammar is not good, sorry!)
In XRD measurements (Fig.1)
In the XRD results, HT and DHT are compared. You mention that the peaks around 2Θ=11.6 and 23.3 disappear due to dehydration (Fig1b) and that DHT is converted (Fig1c). You think that the DHT is still converted and forms a hybrid structure with NIC.
However, in Fig.1d, the NIC peaks are still present (e.g. around 2Θ=13-15 and 25-27, which are very similar to the NIC peaks in Fig1a). I also think that Hybrid is not the basis for the formation. Is NIC-DHT really forming a complex?
In Fig.3, a discussion is necessary.
The nitrogen adsorption curve shows that DHT adsorbs nitrogen gas. This is probably due to adsorption into the dehydrated pores of the DHT structure, which is reasonable. However, the reason why NIC-DHT has the same adsorption curve as HT needs to be discussed. NIC is a poorly soluble drug and the surface barrier would not be large.
It is possible that NIC are adsorbed on the surface of DHT. But isn't it also possible that DHT is adsorbed on the surface of NIC?It is stated earlier that XRD shows that amrophous is retained. This means that the structure of DHT is considered to be disordered. If so, is it reasonable for the story to draw the same curve as HT? Please continue the discussion.
Regarding Fig. 5.
Please indicate the PDI value in the particle size distribution. This information is important in demonstrating the validity of the particle house measurement.
Have you also measured the Zeta potential of these samples? Please suggest this value as it contributes to the stability of the particles.
Reviewer 3 Report
Dear authors,
Please find attached the review report.
Kind regards.
Review report
Manuscript ID: 1188508
Title: Hydrotalcite-Niclosamide-Tween 60 Nanohybrid as Oral Formulation Towards SARS-CoV-2 Viral Infections
Brief summary:
The manuscript aims at developing a Hydrotalcite (DHT)-Niclosamide (NIC)-Tween 60 nanohybrid as oral formulation towards SARS-CoV-2 viral infections. NIC-DHT formulations were physicochemically characterized by Powder X-Ray Diffraction, Fourier Transform Infrared, nitrogen adsorption-desorption isotherms, field-emission scanning electron microscope, and dynamic light scattering. Moreover, the drug content of NIC-DHT-Tween 60 formulations was characterized, and this formulation was used in a in vivo pharmacokinetics study using rats after a single oral administration. The data suggested that the nanostructures are able to improve the bioavailability of NIC upon oral administration.
Broad comments:
The manuscript meets the scope of the journal as it intends to provide a novel drug delivery system to improve the bioavailability of niclosamide, so that this drug can be useful for managing COVID-19 pandemic. The topic of the manuscript is very interesting, and the results may eventually be quite useful. However, I believe some points must be address prior to publication:
- The authors present a comprehensive physicochemical characterization of NIC-DHT formulations, but not of NIC-DHT-Tween 60 formulations. Since the later was under investigation in the PK study, it is crucial to also characterize it, as the adsorption of Tween 60 may alter the properties of the nanosystem.
- The method used to determine the NIC contents in the nanohybrids is not described in the manuscript and the results presented do not display any standard deviation. This way it is not possible to validate the results obtained.
- The PK study is not well described in the materials and method sections. The PK model used to fit the data was not indicated, and the time points selected to collect blood samples did not allow the description of the absorption phase, which hampers the correct determination of tmax and cmax values. Moreover, do the authors have any hypothesis to justify the huge differences observed in the plasma concentration-time curve upon Yomesan and nanohybrid administration?
- Line 42: replace “which necessitates to develop” by “which requires further development”
- Please check throughout the manuscript the blank spaces before ref numbers.
- Line 52: replace “Though” by “Although”
- Line 62: replace “thereby improved” by “thereby improving”
- Lines 108-112: Please clarify the sentence: “And then NIC molecules were loaded on thus prepared DHT to generate 108 the desired NIC-DHT nanohybrid with sufficient drug content (…)”
- Line 282-286: Check the interrogative sentences to correct the grammar errors.
Round 2
Reviewer 3 Report
Dear authors,
Please find attached the review report.
Kind regards.

Round 3
Reviewer 3 Report
Dear authors,
I hope my suggestions were valuable for you to improve the manuscript.
I believe it is now clearer to the readership and the interest of your work is more explicit for all readers.
All the best.